# Theoretical Evaluation of Impact Characteristics of Wavy Graphene Sheets with Disclinations Formed by Origami and Kirigami

**DOI:** 10.3390/nano12030436

**Published:** 2022-01-27

**Authors:** Yoshitada Tomioka, Toshiaki Natsuki, Jin-Xing Shi, Xiao-Wen Lei

**Affiliations:** 1Department of Mechanical Engineering, Graduate School of Engineering, University of Fukui, 3-9-1 Bunkyo, Fukui 910-8507, Japan; tommy.kaeru.1617@gmail.com; 2Institute for Fiber Engineering (IFES), Shinshu University, 3-15-1 Tokida, Ueda 386-8567, Japan; natsuki@shinshu-u.ac.jp; 3Faculty of Textile Science and Technology, Shinshu University, 3-15-1 Tokida, Ueda 386-8567, Japan; 4Department of Production Systems Engineering and Sciences, Komatsu University, Nu 1-3 Shicyomachi, Komatsu 923-8511, Japan; jinxing.shi@komatsu-u.ac.jp; 5Precursory Research for Embryonic Science and Technology (PRESTO), Japan Science and Technology Agency (JST), Saitama 332-0012, Japan

**Keywords:** disclination, wavy graphene sheet, impact characteristics, origami and kirigami, molecular dynamics method, continuum mechanics method

## Abstract

Evaluation of impact characteristics of carbon nanomaterials is very important and helpful for their application in nanoelectromechanical systems (NEMS). Furthermore, disclination lattice defects can generate out-of-plane deformation to control the mechanical behavior of carbon nanomaterials. In this study, we design novel stable wavy graphene sheets (GSs) using a technique based on origami and kirigami to control the exchange of carbon atoms and generate appropriate disclinations. The impact characteristics of these GSs are evaluated using molecular dynamics (MD) simulation, and the accuracy of the simulation results is verified via a theoretical analysis based on continuum mechanics. In the impact tests, the C_60_ fullerene is employed as an impactor, and the effects of the different shapes of wavy GSs with different disclinations, different impact sites on the curved surface, and different impact velocities are examined to investigate the impact characteristics of the wavy GSs. We find that the newly designed wavy GSs increasingly resist the kinetic energy (KE) of the impactor as the disclination density is increased, and the estimated KE propagation patterns are significantly different from those of the ideal GS. Based on their enhanced performance in the impact tests, the wavy GSs possess excellent impact behavior, which should facilitate their potential application as high-impact-resistant components in advanced NEMS.

## 1. Introduction

Impact resistance is an essential evaluation criterion in material design, especially for the application of materials developed for use in aircraft, automobiles, sports products, and machine elements. Since they were first reported [1], graphene sheets (GSs) have attracted enormous interest among scholars globally. Ideal GSs are one-carbon-atom-thick, two-dimensional carbon nanomaterials with hexagonal lattice structures that exhibit excellent mechanical, electrical, chemical, and thermal properties [2,3,4]. In particular, the high-strain-rate behavior [5] of GSs can be exploited for potential applications as high-impact-resistant components in nanoelectromechanical systems (NEMS). The difficulty and complexity of nano-scale impact simulations has meant that impact testing of carbon nanomaterials, such as GSs [6,7,8,9] and carbon nanotubes (CNTs) [10], has been performed based on molecular dynamics (MD) simulation, revealing the high impact resistances of both GSs and CNTs. In addition, a numerical spring–mass model has been adopted to describe the impact behavior of GSs [11,12,13]. Furthermore, theoretical analysis based on continuum mechanics has enabled examination of the transverse impact response of GSs, and the results suggest that the impact velocity can significantly affect the absorption of impact energy by GSs [14,15,16].

However, most of the studies in the literature on the impact characteristics of carbon nanomaterials have focused on ideal carbon nanomaterials with hexagonal lattice structures. During the fabrication of GSs or CNTs, lattice defects, such as dislocations [17], disclinations [18,19,20], and Stone–Wales defects [21], commonly appear owing to high temperatures or external forces. These lattice defects negatively affect the materials, for example by weakening their mechanical properties; however, certain lattice defects have been found to enhance the mechanical and electrical properties of low-dimensional materials [22,23,24,25,26,27,28,29,30,31]. In particular, lattice defects involving heptagons and pentagons play an important role in forming two-dimensional GSs with complex curved surfaces. The introduction of lattice defects to an ideal GS can result in an out-of-plane deformation of the crystal structure because of the spontaneously generated curvature, which transforms the surface from a two-dimensional plane to a three-dimensional curved surface. Because this shape change is a stress relaxation process triggered by the lattice defects, most of the strain energy in the GS is released during the relaxation process. Nelson et al. [32] reported that the surface tension of the curve surface of a 2D crystalline membrane vanished by introducing lattice defects and the out-of-plane deformation could retain a local two-dimensional topology. However, complex stress- and strain-energy distributions are observed depending on the location and types of the lattice defects. Therefore, new functional GSs can be produced with a variety of out-of-plane deformations by appropriately modifying the type, position, and density of lattice defects in GSs. Qin et al. [23] studied the influence of lattice defects on the interlaminar shear strength and in-plane tensile strength of wavy multilayer GSs, revealing that the lattice defects significantly strengthened the mechanical properties of the wavy GSs. Qin at al. [24] also showed that the introduction of lattice defects to GSs yielded wavy GSs with auxetic structures (i.e., structures with negative Poisson’s ratios) and hence also excellent fracture strength and toughness.

Disclinations, which were first observed in 1965 [33], are more complex lattice defects that can cause rotational displacement within multi-scale materials, such as metals [34], carbon structures [35], and strata [36]. To the best of our knowledge, disclinations can induce the most prominent out-of-plane deformations in low-dimensional carbon nanomaterials, especially in GSs [37], and can result in noticeably wavy stable GS structures. Therefore, disclination-induced wavy GSs can also exhibit enhanced impact characteristics.

Buckminsterfullerene (C_60_), discovered in 1985 [38], was used as the impactor in this study because of its spherical shape and structure. C_60_ is a polygon with 60 carbon atoms symmetrically arranged like a soccer ball; the symmetry of this structure assists us to simplify the analytical model and theoretical analysis. In addition, as C_60_ is known to be the smallest fullerene [39], it can be used to impinge on almost every location on wavy GS surfaces, which is convenient for our evaluation of the effects of impact position in this study.

The aims of the study reported herein were to form new wavy GSs by introducing disclination lattice defects and to investigate the impact characteristics of these GSs for potential applications as high-impact-resistant components in advanced nanoelectromechanical systems (NEMS). A technique based on origami and kirigami was adopted to model new wavy GS structures based on controlling complex lattice defect disclinations in ideal GSs to generate out-of-plane deformations. Then, MD simulations of impact tests on ideal and wavy GSs were performed using the fullerene C_60_ as the impactor, and the accuracy of the MD simulation was verified by theoretical analysis of an ideal GS based on continuum mechanics. The influence of the specific wavy GS shapes generated using different disclination gaps, as well as the effects of different points of impact and different impact velocities on the impact resistance, was probed.

## 2. Methods

### 2.1. Simulation Models

Origami and kirigami, traditional Japanese artistic techniques, have been utilized in a wide range of natural science and engineering fields [40]. Miura-ori, conceived by Miura in 1970 [41], involves nonlinear creasing of materials to allow expansion, and folding in the vertical and horizontal directions to improve their strength. We adopted origami and kirigami in this study to form complicated paper models of GSs with disclinations, instead of the complex phase field crystal (PFC) method used for designing curved GS with lattice defects [20,23]. Positive and negative disclinations were introduced by removing and inserting wedge shapes from/into an ideal planar GS, respectively. The details of the proposed origami- and kirigami-based technique for forming wavy GSs with disclinations are presented in Figure 1a. Based on the paper models constructed using the origami–kirigami technique, atomic models of the disclination-incorporated wavy GSs were created, with different types of disclinations generating different out-of-plane deformations and yielding differently shaped GSs. A four-membered ring in the GSs was defined as an isolated positive disclination that forms a conical curved surface, whereas an eight-membered ring in the GSs was defined as an isolated negative disclination that forms a saddle-shaped curved surface, as shown in Figure 1b. Thus, each repeating unit of the wavy GSs was constructed as a disclination quadrupole combining two pairs of the four- and eight-membered rings, i.e., four isolated disclinations. Observing the structure along the *z*-axis from the positive direction, the two eight-membered rings are arranged in the upper right and lower left regions, and the two four-membered rings are arranged in the lower right and upper left regions. The four isolated disclinations connected in this manner were regarded as one basic wavy unit, and multiple units were connected to create a unique wavy GS model constructed as a network of disclinations. Although an ideal GS has a stable structure with a planar shape, the introduction of disclinations formed a GS with an inherently curved surface, as shown in Figure 2. Furthermore, the wavy GS with an assembly of disclinations exhibited greater stability than the ideal GS, with improvements in the impact characteristics of the GS that might possibly be attributed to the corrugated structure.

As shown in Figure 2, the impact tests involved impinging the fullerene C_60_ (mass, 1.197 zg; diameter, 6 Å) on a GS with dimensions of 120 Å × 120 Å. Moreover, all four edges of the target GSs were fixed during the impact test. Impact tests were performed on an ideal GS and four types of wavy GSs (named as wavy1, wavy2, wavy3, and wavy4) with different disclination densities, as shown in Table 1. The central position in the z direction of each GS model was set to 0 Å. Because C_60_ is initially positioned above the GS, it was necessary to place it at a sufficiently large distance to avoid the influence of the van der Waals (vdW) interaction forces between the carbon atoms of C_60_ and the GS (20–30 Å) [42]; therefore, in this work, the height of C_60_ was set to 30 Å. As detailed in Table 1, l was the distance between adjacent upward-facing four-membered rings (Figure 2), and the disclination density was defined as the number of disclinations per unit area. After performing the impact simulation with different types of GSs, the wavy4 model with *l* = 31.79 Å was selected for further analysis using different positions of impact and impact velocities. With respect to the points of impact, the lowest and highest points (z positions with the greatest magnitudes) nearest to the center of the wavy GS model were defined as A0 and A4, respectively; moreover, three evenly spaced points between A0 and A4 were defined as A1, A2, and A3 (black circles in Figure 2).

### 2.2. MD Analysis Conditions

The impact tests were performed using the Large-scale Atomic/Molecular Massively Parallel Simulator (LAMMPS) package [43], and atomic-scale visualization was performed using the Open Visualization Tool (OVITO). The adaptive intermolecular reactive empirical bond order (AIREBO) [44] force field (Equation (1)), which has been extensively used for probing the mechanical properties of carbon nanomaterials in general, was employed for all the intra- and inter-molecular interactions.
(1)$$\begin{array}{c} E=\frac{1}{2}\sum_{i}\sum_{j\neq i}\left[E_{ij}^{\mathrm{REBO}}+E_{ij}^{\mathrm{LJ}}+\sum_{k\neq i,j}^{}\sum_{l\neq i,j,k}^{}E_{kijl}^{\mathrm{tors}}\right] \end{array}$$
where $E_{ij}^{\mathrm{REBO}}$ is the REBO potential of a covalent bond, $E_{ij}^{\mathrm{LJ}}$ is the Lennard–Jones (LJ) potential of a non-covalent bond, and $E_{kijl}^{\mathrm{tors}}$ is tortional potential that depends on the dihedral angle. Because the AIREBO potential has been applied for calculating dihedral angles, it was a suitable choice for use in evaluating the curved surface model characteristics of the constructed GSs. In addition, because the vdW interaction forces cannot be ignored, the interactions between the carbon atoms in C_60_ and the GSs were considered. The cut-off distance in the potential was set to 1.7 Å to maintain the smoothness of the wavy GS structures and avoid the scattering of atoms during the impact tests. A time step of 1.0 fs and a temperature of 5 K were employed in all the simulations. Prior to the impact tests with C_60_, the target GSs and C_60_ were fully relaxed and equilibrated at 5 K and a pressure of 0 atm in an isothermal–isobaric ensemble (NPT) for 5 ps. The relaxation was conducted using the conjugate gradient (CG) method, and the accuracy (Δ*E*) was set to 10^−20^. Moreover, all the impact tests were conducted using the NVE ensemble, i.e., Newton’s equation.

### 2.3. Continuum Mechanics Method

A theoretical analysis of an ideal GS based on continuum mechanics was performed with the aim of verifying the accuracy of our MD simulations. According to the continuum mechanics method, an ideal GS can be simulated using a continuum plate model; hence, the governing equation of motion for a rectangular GS can be written as follows:(2)$$D\frac{\partial^{4}z\left(x,y,t\right)}{\partial x^{4}}+2D\frac{\partial^{4}z\left(x,t\right)}{\partial x^{2}\partial y^{2}}+D\frac{\partial^{4}z\left(x,t\right)}{\partial y^{4}}+\rho h\frac{\partial^{2}z\left(x,t\right)}{\partial t^{2}}=q\left(x,y,t\right),\;$$
where z(x,y,t) is the flexural deflection of the rectangular GS, *t* is the time, q(x,y,t) is the transverse load acting on the GS plate, and *ρ* is the density of the GS. *D* is the flexural rigidity of the GS plate, which is expressed as follows:(3)$$\begin{array}{c} D=\frac{Eh^{3}}{12\left(1-\nu^{2}\right)}, \end{array}$$
where *E* is the elastic modulus of the GS, *h* is the thickness of the GS, and *ν* is Poisson’s ratio for the GS.

All edges of the rectangular GS are simply supported; therefore, the displacement and load function could be represented as follows:(4)$$\begin{array}{c} z\left(x,y,t\right)=\sum_{m=1}^{\infty }\sum_{n=1}^{\infty }Z_{mn}\left(t\right)\sin\frac{m\pi x}{L_{a}}\sin\frac{n\pi y}{L_{b}}, \end{array}$$
(5)$$\begin{array}{c} q\left(x,y,t\right)=\sum_{m=1}^{\infty }\sum_{n=1}^{\infty }Q_{mn}\left(t\right)\sin\frac{m\pi x}{L_{a}}\sin\frac{n\pi y}{L_{b}}, \end{array}$$
where $Z_{mn}\left(t\right)$ is the time-dependent coefficient, and $Q_{mn}\left(t\right)$ is a term corresponding to the Fourier series expansion. *L*_a_ and *L*_b_ (both equal to 120 Å) are the side lengths of the GSs.

A concentrated load, F(t), is assumed to be located at the (ξ,η) position. Therefore,
(6)$$\begin{array}{c} Q_{mn}\left(t\right)=\frac{4F\left(t\right)}{L_{a}L_{b}}\sin\frac{m\pi \xi }{L_{a}}\sin\frac{n\pi \eta }{L_{b}}. \end{array}$$

Substituting Equations (3)–(6) into Equation (2) resulted in the following expression:(7)$$\begin{array}{c} {\ddot{Z}}_{j,mn}+\omega_{j,mn}^{2}Z_{j,mn}=\frac{4F\left(t\right)}{\rho hL_{a}L_{b}}\sin\frac{m\pi \xi }{L_{a}}\sin\frac{n\pi \eta }{L_{b}}, \end{array}$$
where *j* = 1,2, and
(8)$$\begin{array}{c} \omega_{mn}^{2}=\frac{\pi^{4}D}{\rho h}\left[{\left(\frac{m}{L_{a}}\right)}^{4}+2{\left(\frac{m}{L_{a}}\right)}^{2}{\left(\frac{n}{L_{b}}\right)}^{2}+{\left(\frac{n}{L_{b}}\right)}^{4}\right]. \end{array}$$

The solution to Equation (7) can be expressed as
(9)$$Z\left(t\right)=\frac{4}{\rho hL_{a}L_{b}\omega_{mn}}\sin\frac{m\pi \xi }{L_{a}}\sin\frac{n\pi \eta }{L_{b}}\int_{0}^{t}F\left(\tau \right)\sin\omega_{mn}\left(t-\tau \right)d\tau .\;$$

The deflection at an impact position (ξ,η) located in the middle of the GS plate can be expressed as
(10)$$w\left(\frac{L_{a}}{2},\frac{L_{b}}{2},t\right)=\frac{4}{\rho hL_{a}L_{b}}\underset{m=1,3,5,\cdots \;}{\sum }\underset{n=1,3,5,\cdots }{\sum }\int_{0}^{t}F\left(\tau \right)\sin\omega_{mn}\left(t-\tau \right)d\tau .\;$$

Therefore, the contact deformation of the rectangular GS plate can be written
(11)$$\alpha \left(t\right)=w\left(t\right)-\frac{4}{\rho hL_{a}L_{b}}\underset{m=1,3,5,\cdots \;}{\sum }\underset{n=1,3,5,\cdots }{\sum }\int_{0}^{t}F\left(\tau \right)\;\sin\omega_{mn}\left(t-\tau \right)d\tau ,\;$$
where w(t), as expressed in Equation (12), gives the displacement of the impact mass.
(12)$$\begin{array}{c} w\left(t\right)=v_{F}^{0}t-\frac{1}{m_{0}}\int_{0}^{t}F\left(\tau \right)\left(t-\tau \right)d\tau , \end{array}$$
where $m_{0}$ and $v_{F}^{0}$ are the mass and initial velocity of the impactor C_60_, respectively.

The Hertz law of contact [45] was employed to express the relationship between the contact force, F(t), and deformation, α(t), as follows:(13)$$\begin{array}{c} F\left(t\right)=K\alpha {\left(t\right)}^{3/2}, \end{array}$$
where K, the Hertzian contact stiffness, was determined using the radius of the impactor and the stiffness values of the impactor and GS plate. The energy Ψ(*t*) absorbed by the GS plate within time *t* can be expressed in term of the impact force *F* acting on the GS plate and the displacement *w* of the point of impact:(14)$$\begin{array}{c} \Psi \left(t\right)=\int_{0}^{t}F\left(\tau \right)dw=\int_{0}^{t}F\left(\tau \right)\dot{w}\left(\tau \right)d\tau . \end{array}$$

## 3. Results and Discussion

### 3.1. Comparison of Ideal GS Impact Tests via MD Simulation and Continuum Mechanics Method

Impact tests of an ideal GS using MD simulation and the continuum mechanics method were first compared. Figure 3 shows the calculated energies for the impact of an ideal GS with C_60_ at an initial impact velocity of 10 Å/ps, along with the total kinetic energies (KEs) and potential energies (PEs) values of GS and C_60_, respectively, which suggest that the KE of the impactor can be absorbed as the KE and PE of GS. For GS, the impact-induced deformations were not completely preserved, and it returned to its initial shape, resulting in as absorption of the KE of the impactor; the KE of GS is a little higher than the KE of the impactor due to the newly generated negative PE of GS. For C_60_, because it was considered as a rigid body, the KE of C_60_ after impact became completely 0.0 eV. In addition, the theoretical value of the initial KE of fullerene, $(m_{F}{\left(v_{F}^{0}\right)}^{2})/2$ = 3.75 eV, was almost equal to total energy, which was depicted in Figure 3 (purple line) to facilitate this comparison.

In the MD simulation, the initial distance between C_60_ and the ideal GS was set as 30 Å to eliminate the influence of the vdW interaction forces; therefore, a period of descent of the C_60_ prior to its impact on the GS was considered. The KE of the GS during this period was a nonzero constant value prior to the start of the impact. The atoms in the GS were presumed to sensitively react to slight changes prior to impact and continue to experience some movement after the impact. When C_60_ impinged on the GS, the KE of the GS considerably increased with increasing impact time. However, the theoretical results obtained using the continuum mechanics method showed that the KE of the GS increased from zero and converged to a value a little higher than the initial KE of the impinging C_60_. The KE of the GS prior to impact was zero in the theoretical calculations because the minimal motions of the atoms in the GS were neglected. A comparison of the post-impact KEs of the GS revealed reasonable agreement between the KEs obtained using the two methods.

In particular, the results satisfy the law of conservation of energy, indicating the suitability of the employed MD simulation for investigating the impact tests. Similar MD simulation impact tests were performed on the wavy GSs with disclinations, and the effects of disclination density, KE distribution, site of impact, and impact velocity were examined.

### 3.2. Effects of Disclination Density

Impact simulations on the ideal and four wavy GSs with different disclination densities (Table 1) were conducted using $v_{F}^{0}$ = 10 Å/ps to analyze the impact characteristics, as shown in Figure 4. For the ideal and wavy GSs, the points of impact were at the center and the A4 site near the center, respectively (Figure 2). Figure 4a shows the KEs of each GS obtained during the impact tests. The A4 location corresponds to the tip of a protruding portion of the wavy GSs, and represents the region with the greatest *z*-coordinate value. Therefore, A4 is guaranteed to be in contact with any impactor for at least some time during the collision, even if the impactor is larger than C_60_ or has a flat tip. In addition, the pre-impact KEs of the ideal and wavy GSs are almost the same, presumably because the numbers of carbon atoms in the models were composed of approximately the same 10,000 carbon atoms; the Kes per carbon atom of the ideal and wavy GSs were approximately equal to each other. A difference can be observed in the impact time at which the KE of each GS began to increase because of the varying heights of the A4 sites of the different wavy models. Therefore, wavy4, which has the highest A4 site, was impacted the earliest, followed by wavy3; the ideal GS exhibited the longest impact time. At the end of each impact test, the KE of all the GS converged to approximately the same value, i.e., the initial KE of the C_60_ impactor.

During the impact simulation, the wavy1 model exhibited higher KE values than the other models. However, in terms of the change in KE immediately before and after the impact, that is, the impact-induced change in KE, the ideal and wavy1 GSs models displayed similar behavior, with the other models showing smaller changes. In essence, the ideal GS exhibited a greater capacity to absorb the KE of the impactor, whereas the wavy4 GS absorbed the least KE from the impactor, as shown in Figure 4b. Δ*E* is the amount of KE change of GSs before and after being impacted by C_60_, as shown in Figure 4a. The wavy GSs have high resistance to impact. This behavior can also be observed by comparing the KEs of C_60_ during the collisions, as shown in Figure 4c. The C_60_ that collided with the ideal GS lost almost all of its KE, whereas the C_60_ units that collided with the wavy2, wavy3, and wavy4 GSs retained KEs of approximately half of their original value after impact. Compared to the ideal GS, the wavy GSs were better able to resist the KE of the impactor.

### 3.3. Distribution of KE

The impact-related changes to the various GS model structures were elucidated using the KE distributions of the GSs at representative durations of 0.0, 0.5, 1.0, and 1.5 ps (Figure 5). Although the initial sizes of the five models are similar, different KE distributions are exhibited because of the different out-of-plane deformations produced by the introduced disclinations. For the ideal, disclination-free GS, the KE distribution expanded as a concentric circle from the central point of impact and was eventually distributed throughout the GS. The wavy1 model with the highest disclination density showed the most extended KE distribution among the four wavy GSs. However, the KE distribution is circular at 0.5 ps and spread in a cross-shaped pattern along the *x* and *y* directions, with negligible propagation in the diagonal direction. The wavy2, wavy3, and wavy4 models almost did not absorb KE from the fullerenes, which is consistent with the analysis presented in the previous section; therefore, no noticeable trends were observed in their KE distributions. Videos of the KE distribution during the impact tests obtained via MD simulation, provided as Appendix A, clearly illustrate the spread of the KE distribution. Therefore, in the ideal GS, the impact-induced KE spread concentrically and was ultimately dispersed throughout the GS, whereas in the wavy GSs, which absorbed similar amounts of KEs as the ideal GS, the KE propagation was limited in terms of direction and range.

### 3.4. Effects of Impact Location

Different impact characteristics, such as energy conversion and GS deformation, were obtained for the collisions between C_60_ and different locations within the wavy GSs. The wavy4 model, which had the lowest disclination density, was selected as a representative GS to reveal the effects of impact location. The wavy4 model was impacted with C_60_ at $v_{F}^{0}$ = 10 Å/ps at five locations (A0–A4; Figure 2), and the results are shown in Figure 6. The moments of impact at the different sites occurred at different times because of the different distances between each impact location and C_60_. After the impact, the KEs of the GS impacted at the A0, A1, A2, and A3 sites converged to higher KE values. The A0 site corresponds to the lowest point on the surface (most negative *z*-coordinate value) of the inverted cone in the wavy GSs; therefore, C_60_ collides with the A0 site in a manner similar to the dropping of a mass into a saucer, i.e., with minimal bouncing. After the first impact of C_60_ on the GS at the A3 site, it bounces and impacts the GS again at a different location, which results in two energy changes in the curve corresponding to A3 prior to convergence. In the scenarios involving the A0, A1, and A2 sites, the post-impact KEs increase only slightly over time and nearly converged to the converged KE of location A3; however, the KE corresponding to location A4 converged to a value approximately half that of the converged KE of the other impact sites. This is because the collision of C_60_ with site A4, which is the highest point on the wavy GS surface (most positive *z*-coordinate value), resulted in C_60_ bouncing directly off the GS and not bouncing back onto it, unlike the behavior at the other impact sites. However, because location A4 is the highest point on the wavy GS surface, it can be presumed that this is the most probable location for impact, especially when the diameter of the impactor is considerably larger than that of C_60_. The KE absorption at A4 is lower than that at the other locations. Therefore, when a collision occurs at A4, the wavy GS can presumably retain the impact-induced shape change, that is, it can absorb more KE. By contrast, the protruding portion of the surface cannot easily resist the deformation, which means that there is little KE absorption. Therefore, KE absorption by the wavy GSs is strongly dependent on the site of impact. The wavy structure can be approximately classified into protruding areas and other areas, and we established that KE absorption by the wavy GSs was greater for collisions with surface depressions.

### 3.5. Effects of Impact Velocity

The wavy GSs exhibited reduced Young’s moduli and bending moments compared to those exhibited by the ideal GS. The wavy GSs can be considered as consisting of deformable shells. To investigate the effects of impactor velocity on the KEs of the GSs, the wavy1 model was impacted at site A4 by C_60_ at velocities, $v_{F}^{0}$, of 10, 20, 30, 40, 50, and 60 Å/ps. As shown in Figure 7, the pre-impact KEs of all the GSs are similar. However, all the KEs of the GSs converged to different constant values. For comparison, short lines colored to match the corresponding KE curve have been included on the right in Figure 7; these lines indicate the values of $(m_{F}{\left(v_{F}^{0}\right)}^{2})/2$, which increase with impactor velocity. However, the converging value of the KE is close to $(m_{F}{\left(v_{F}^{0}\right)}^{2})/2$ only at $v_{F}^{0}$ = 10 Å/ps, which can be explained by considering the conservation of energy. With increasing impact velocity, the difference between the value of $(m_{F}{\left(v_{F}^{0}\right)}^{2})/2$ and the corresponding converging KE increased. This is because of the microvibrations of the C_60_ colliding with the wavy GS at low impact velocities, and the rebounding of the C_60_ from the GS at higher velocities; the wavy1 GS was ruptured when C_60_ collided with the GS at $v_{F}^{0}$ = 60 Å/ps. In particular, at higher impact velocities, the deformation of the GS was considerably greater than that achieved at lower impact velocities. The deformation energies of the GSs with different impact velocities are shown in the inset of Figure 7, confirming that the KEs of the GSs impacted by C_60_ at higher velocities are retained at higher velocities.

## 4. Conclusions

In this work, new stable wavy GSs were generated by appropriately introducing disclinations into an ideal GS using a technique based on origami and kirigami. The impact characteristics of wavy GSs with incorporated disclinations were investigated using MD simulations, and the accuracy of these was verified using a continuum mechanics method. The disclination density significantly affected the impact characteristics of the wavy GSs and even determined their energy absorption and dispersion rates. Because disclinations can produce prominent out-of-plane deformations in GSs, they can reduce the spread of post-impact effects. Consequently, the KE distribution in the ideal GS was an expanding circle, whereas that in the wavy GS exhibited cross-shaped (local) propagation, indicating that the wavy GS structures offer control over the range of KE propagation. By comparing the impact results obtained at different points of impact, the C_60_ impactor was found to undergo various numbers of bounces and possess varying KE absorption, owing to the nonuniformity of the disclination-induced out-of-plane deformations in GSs. Nonetheless, with respect to the impact velocity, the absorption ratio of the target GS decreased at velocities higher than $v_{F}^{0}$ = 10 Å/ps. Therefore, the results of these impact tests indicate that these new wavy GS structures possess improved impact characteristics with respect to ideal GSs, which should promote their application as high-impact-resistant components in advanced NEMS.

## Figures and Tables

**Figure 1 nanomaterials-12-00436-f001:**
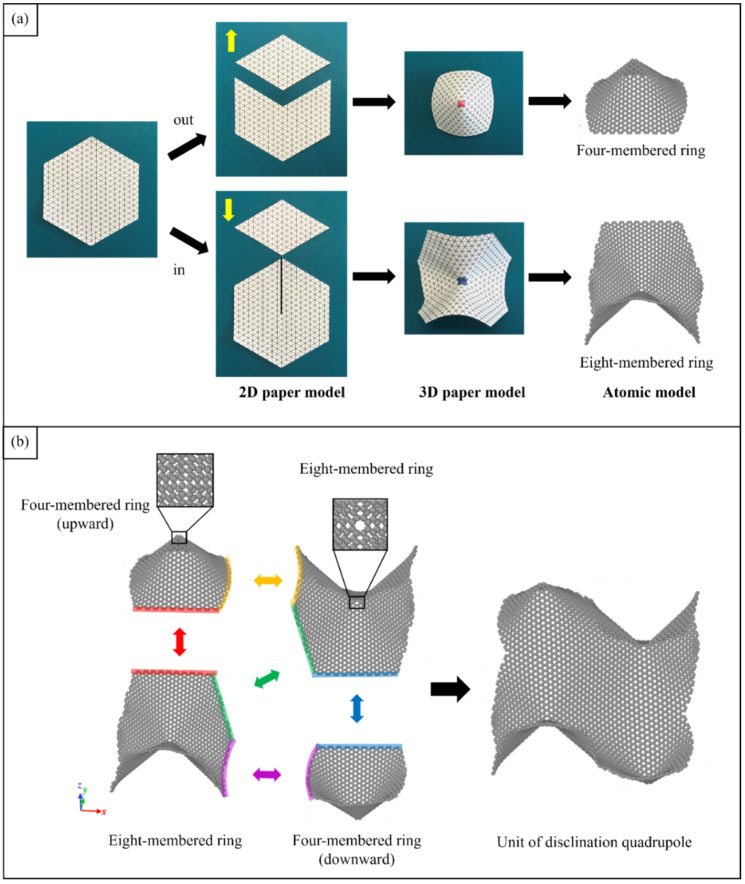
(**a**) Generation of disclination models; (**b**) Atomic model of the disclination quadrupole unit composed of isolated disclinations of two pairs of four- and eight-membered rings.

**Figure 2 nanomaterials-12-00436-f002:**
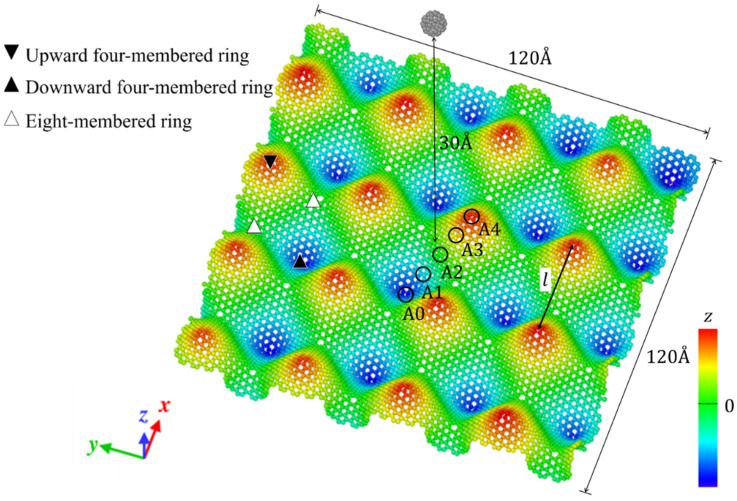
Illustration of the simulation model with detailed depiction of the wavy GS.

**Figure 3 nanomaterials-12-00436-f003:**
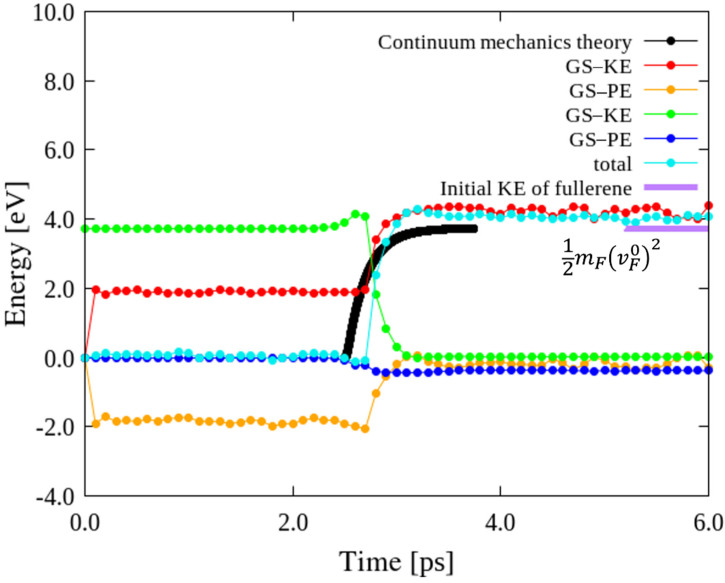
Comparison of energies of an ideal GS impacted by C_60_ estimated using MD simulation and a method based on continuum mechanics. The thick purple line at the top at 3.75 eV represents the initial KE of the C_60_ with an initial velocity $v_{F}^{0}$ = 10 Å/ps. KE, PE, and total energies of the different types of GSs and C_60_ during the impact tests.

**Figure 4 nanomaterials-12-00436-f004:**
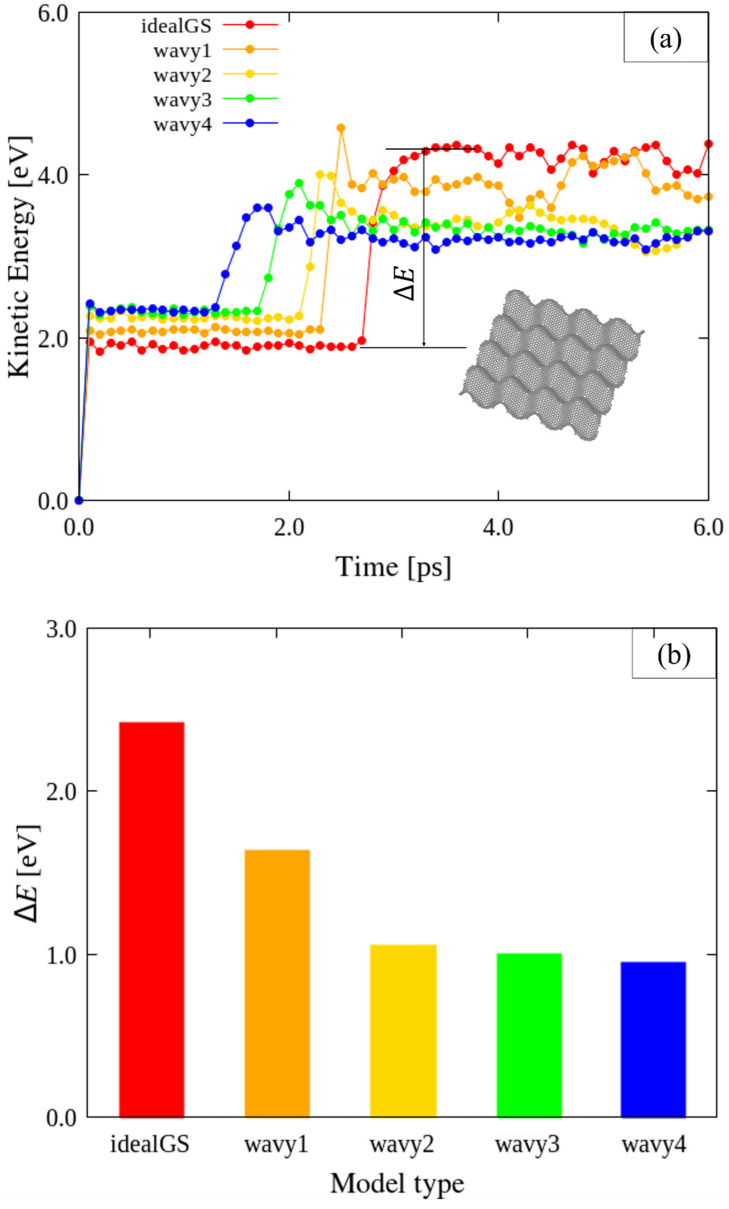
Relationship between impact energy and time during collisions between GSs and C_60_ with an initial velocity $v_{F}^{0}$ of 10 Å/ps. (**a**) Comparison of KEs of the different types of GSs impacted by C_60_. (**b**) Comparison of Δ*E* for the different types of GSs impacted by C_60_. (**c**) Comparison of KEs of C_60_ impacting the different types of GSs.

**Figure 5 nanomaterials-12-00436-f005:**
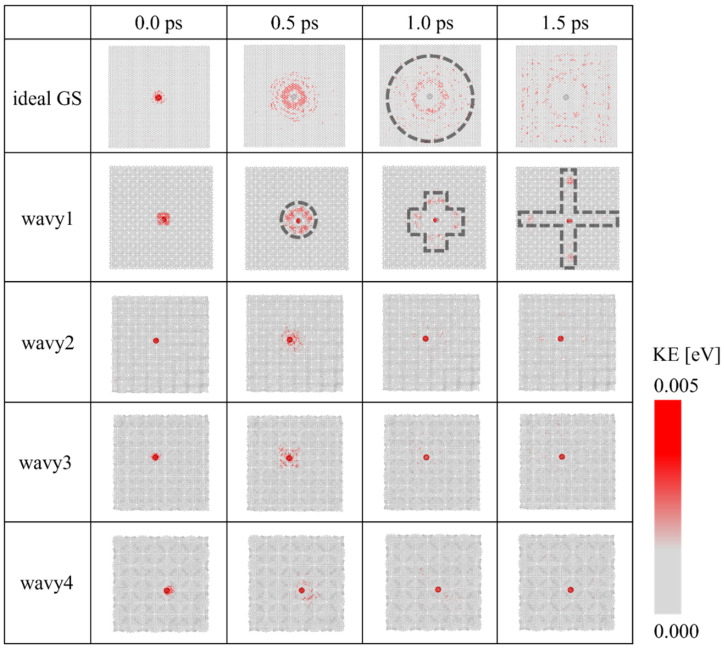
Time evolutions of KE distributions of the GS models impacted by C_60_ with an initial velocity $v_{F}^{0}$ of 10 Å/ps. The distributions at 0, 0.5, 1.0, and 1.5 ps were computed based on the moment of impact corresponding to 0 ps.

**Figure 6 nanomaterials-12-00436-f006:**
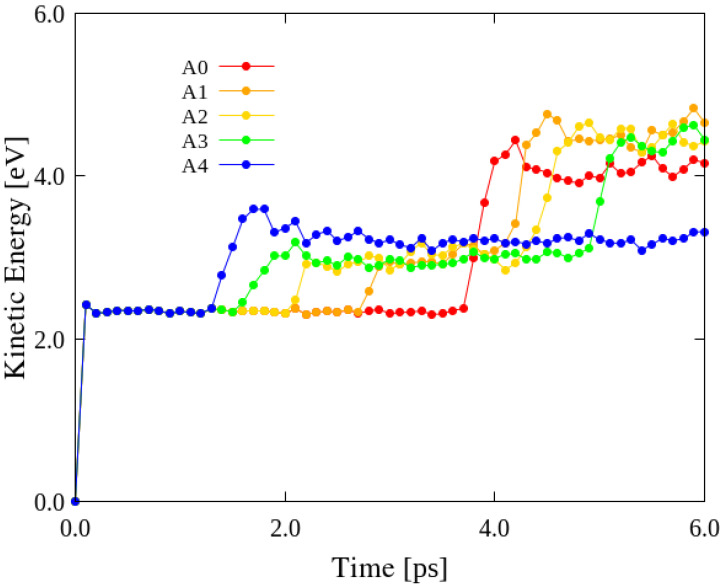
Comparison of KEs of the wavy4 GS model impacted by C_60_ at $v_{F}^{0}$ = 10 Å/ps at the A0, A1, A2, A3, and A4 sites.

**Figure 7 nanomaterials-12-00436-f007:**
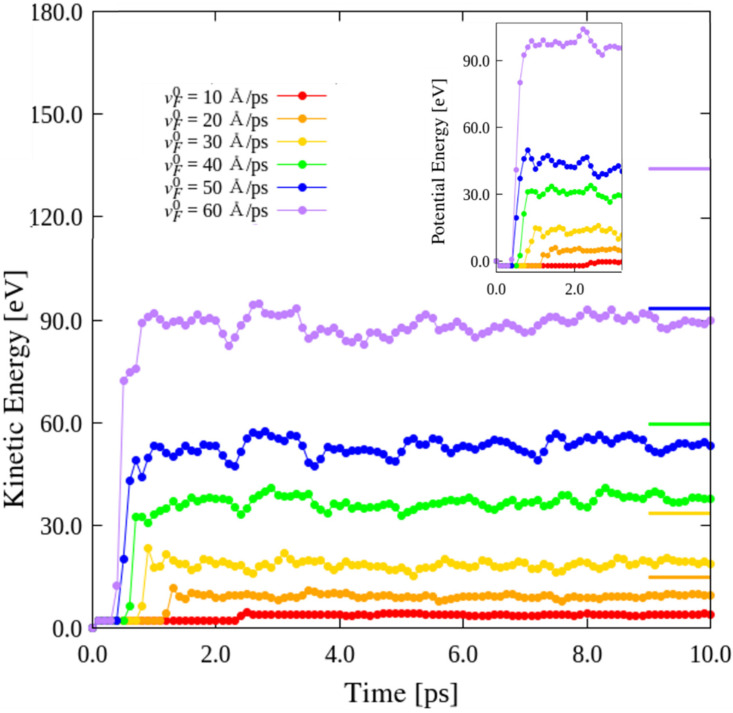
Comparison of KEs of wavy1 GS impacted by C_60_ with impact velocities $v_{F}^{0}$ = 10, 20, 30, 40, 50, and 60 Å/ps at A4.

**Table 1 nanomaterials-12-00436-t001:** Parameters of the four types of wavy GSs.

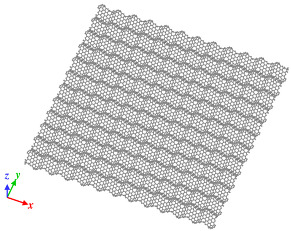 Model of wavy1 GS, *l* is 11.42 Å. Disclination density is 31.30 × 10^−3^/Å^2^.	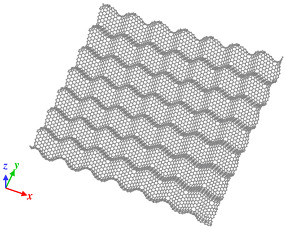 Model of wavy2 GS, *l* is 18.06 Å. Disclination density is 11.40 × 10^−3^/Å^2^.
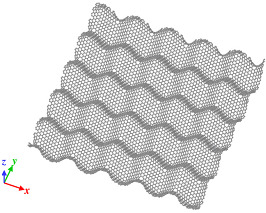 Model of wavy3 GS, *l* is 25.38 Å. Disclination density is 6.29 × 10^−3^/Å^2^.	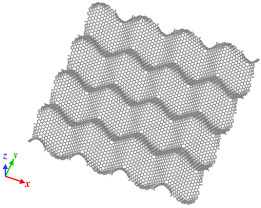 Model of wavy4 GS, *l* is 31.79 Å. Disclination density is 3.96 × 10^−3^/Å^2^.

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
