# Peer review of "Theoretical Evaluation of Impact Characteristics of Wavy Graphene Sheets with Disclinations Formed by Origami and Kirigami"

_nanomaterials, 2022, doi:10.3390/nano12030436_

Round 1

Reviewer 1 Report

In this paper, the authors design novel stable wavy graphene sheets (GSs) using a technique based on origami and kirigami to control the exchange of carbon atoms and generate appropriate disclinations. The impact characteristics of the GSs are investigated using MD simulations, and the accuracy of the simulation results is verified using a continuum mechanics method. It is found from the impact tests that the wavy GSs possess excellent impact behavior, which should facilitate their potential application as high-impact-resistant components in advanced NEMS.

The work is interesting and novel, but the authors need to address the following comments before its publication in the journal.

  1. The accuracy of MD simulations is verified by using the theory of continuum mechanics only in the case of ideal GSs. However, for wavy GSs, it is necessary to demonstrate whether the MD simulations conform to the analytical solutions by the continuum mechanics model.
  2. According to the displacement function form in Eq. (4), it seems more reasonable to change the expression that all edges of the rectangular GS are fixed into that all edges of the rectangular GS are simply supported.
  3. The Q_mn in Eq. (5) should be changed to Q_mn (t) since it is time dependent.
  4. The authors should add a legend in Fig. 3 to illustrate the meaning of the “thick purple line” more intuitively.

Author Response

We appreciate the encouraging, critical and constructive comments on this manuscript by the reviewer. The comments are very useful for improving the manuscript. We have taken them fully into account in the revised manuscript and believe that the comments and suggestions should enhance the scientific value of the revised manuscript by many folds. The responses are given in attachment.

Reviewer 2 Report

This manuscript addresses the MD modeling of impact of C60 carbon structures onto graphene sheets (GS) with different levels of waviness. The waviness in the GS were built based on origami methods. The results show different responses of the GS to the impact, based on the level of waviness. Here are my comments:

  1. The authors repeatedly refer to their “impact experiments”. However, there are no laboratory experiments performed in this study. The MD simulations should be referred to as “impact simulations”
  2. In relation to the results shown in Figure 4, I think it would be most helpful if the authors included the changes in Temperature to help further understand the influence of waviness on the changes in kinetic energy. Because the authors used LAMMPS and the NVE ensemble, they should also have instantaneous temperature values, which will show how the increases energy absorption of the GS manifests itself with different changes in temperature.

Author Response

(The authors gave the same response as above.)

Round 2

Reviewer 1 Report

Publication of the manuscript is recommended.